# Truck platooning reshapes greenhouse gas emissions of the integrated vehicle-road infrastructure system

Huailei Cheng [1,2], Yuhong Wang [1] ✉, Dan Chong[3], Chao Xia[4], Lijun Sun [2] ✉, Jenny Liu[5], Kun Gao[6], Ruikang Yang[2] & Tian Jin[2]

Reducing greenhouse gas emissions has turned into a pillar of climate change mitigation. Truck platooning is proposed as a strategy to lower emissions from vehicles on roads. However, the potential interactive impacts of this technology on road infrastructure emissions remain unclear. Here, we evaluate the decarbonization effects of truck platooning on the integrated vehicle-road system at a large-scale road network level, spanning 1457 road sections across North America. We show that truck platooning decreases emissions induced by truck operations, but it degrades faster the durability of road infrastructure and leads to a 27.9% rise in road emissions due to more frequent maintenance work. Overall, truck platooning results in a 5.1% emission reduction of the integrated vehicle-road system. In contrast to the benefits of emission reduction, truck platooning leads to additional financial burdens on car users and transportation agencies, calling for the consideration of tradeoffs between emissions and costs and between agencies and users. Our research provides insights into the potential applications of truck platooning to mitigate climate change.

Anthropogenic Greenhouse Gas (GHG) emissions are rising across all major industrial sectors[1]. Increased GHG emissions result in climate changes such as global warming, extreme weather, land degradation, and ocean current variation[2–4]. Cutting down GHG emissions has become a necessary step toward combatting global climate change. The transport sector is estimated to contribute 23% (8.7 Gt $CO_2$-eq) of global energy-related emissions[1], with fossil fuel consumption by vehicles being the primary source[5,6]. For instance, the US EPA (Environmental Protection Agency) reports that emissions from road vehicles account for 22% of total US GHG emissions[7]. Therefore, developing effective decarbonization strategies and technologies in the road transport sector holds great potential for GHG reductions.

One recently proposed strategy to reduce vehicle-related GHG emissions is truck platooning[8,9]. Truck platooning resembles the operation mode of the train, with trailing trucks following closely with a heading truck to reduce air resistance and thus fuel consumption. The latest aerodynamic simulations and field observations have revealed that truck platooning improves the fuel economy of trucks and reduces vehicle-generated GHG emissions[10–13]. However, vehicles closely interact with road infrastructure in a transportation system. In particular, truck platooning reduces the loading interval between two consecutive truck loads, likely hindering the self-healing of the road pavement layer and damaging road durability as compared to normal truck operations[14–18]. Consequently, platooning trucks can increase the demands on road maintenance work (e.g., crack sealing, patching,

[1]Department of Civil and Environmental Engineering, The Hong Kong Polytechnic University, Hong Kong SAR, China. [2]The Key Laboratory of Road and Traffic Engineering of Ministry of Education, Tongji University, Shanghai, China. [3]School of Management, Shanghai University, Shanghai, China. [4]Shanghai Automotive Wind Tunnel Center, Tongji University, Shanghai, China. [5]Department of Civil, Architectural and Environmental Engineering, Missouri University of Science and Technology, Rolla, MO, USA. [6]Department of Architecture and Civil Engineering, Chalmers University of Technology, Gothenburg, Sweden. ✉e-mail: yuhong.wang@polyu.edu.hk; ljsun@tongji.edu.cn

milling, and overlay) after initial construction and shorten the pavement's service life. Road maintenance work after initial construction produces GHG emissions not only through the work itself but also from lost efficiency of vehicle operations (e.g., traffic congestion)[19–22]. The additional emissions from road maintenance reshape the carbon footprints of the transportation system, making the net decarbonization benefits of truck platooning uncertain. Additionally, existing studies on the benefits of truck platooning are limited to project-level investigations covering only a few traffic and environment scenarios. Considerable uncertainty remains in the effectiveness of truck platooning if this technology is introduced to large-scale road networks.

This study examined the effects of truck platooning on the GHG emissions (i.e., $CO_2$-eq) of the integrated vehicle-road infrastructure system at a network level. A total of 1457 road sections across North America were used for assessments for two reasons. Firstly, per capita vehicle ownership, road travel mileage, and road travel-related GHG emission in North America rank the top in the world. Therefore, any potential decarbonization benefits in this region can lead to non-negligible impacts on global GHG reduction. Secondly, an extensive road information database has been developed for this region by the Long-term Pavement Performance (LTPP) program[23]. The LTPP database provides essential data for emission analysis, including road locations, structure and material properties, traffic volumes, climate zones, road performance, and maintenance work (see Supplementary Method 1, Supplementary Data 1–5). With those data, we calculated and assessed carbon footprint variations generated by truck platooning (assumed) vs. non-platooning (baseline) alternatives via a developed framework (see Supplementary Fig. 1). The results shed light on how truck platooning affects GHG emissions of the vehicle-road infrastructure system.

## Results

### Truck platooning decreases GHG emissions of vehicles

Truck platooning impacts the spatial-temporal distributions of vehicles on the road[24–29]. To investigate this, we employed a traffic flow simulation tool to analyze vehicle behaviors under truck-platooning mode and those under normal operation mode (see Supplementary Method 2). The outputs from traffic flow simulations were combined with the fuel consumption and emission models we developed (See Supplementary Methods 3 and 5) to evaluate the impacts of truck platooning on emissions from vehicles. Truck platooning primarily aims to save fuel and reduce GHG emissions from participating trucks. Since vehicles on roads include both trucks and passenger cars, we first examined the relative emission contributions from these two types of vehicles on 1457 road sections. We find that emissions from trucks (single trucks & combo trucks) contribute to a noticeable portion of

the overall vehicle emissions (emissions from both trucks and cars) under the normal operation mode (Fig. 1a). Emissions from trucks, although varying with road sections, account for 68.5% of the total emissions on average. Therefore, if platooning can help reduce truck emissions, it will help lower overall vehicle emissions.

We then calculated the emissions from trucks under the platooning mode and compared them with those under the normal operation mode. We confirm, as expected, that trucks operating under the platooning mode produce fewer emissions than those under the normal mode. This is because platooning reduces air resistance and thus improves fuel economy. We characterized the emission reduction efficiency of platooning by calculating the decreasing rate, defined as the percentage reduction in truck emissions when the platooning mode is applied. We observe that the decreasing rate is affected by truck type (Fig. 1b): for single trucks, it ranges from 7.4% to 11.3% at different road sections with an average value of 9.8%; for combo trucks, it ranges from 6.6% and 20.3% with an average value of 12.8%. Overall, the average decreasing rate is 11.2% for all trucks. As for the emissions from passenger cars, they are regarded as unaffected by truck platooning. This is because road sections undergo free traffic flows at normal operation periods (i.e., without maintenance activity) according to our traffic simulations. Thus, the platooning of trucks causes negligible disruptions to passenger car operations.

If emissions from passenger cars are also added, the benefits of reducing emissions through truck platooning become less significant (Fig. 1c). The platooning-caused decreasing rate of vehicle emissions ranges from 0.6% to more than 10.0% for different road sections. The decreasing rate of emissions is affected by the proportion of trucks on the roads (Fig. 1a, c). As expected, more trucks on a road result in higher emission savings if platooning is applied. The mapping of decreasing rates also exhibits a valley area in the east coastal region of the U.S., where urban roads are mainly used to serve passenger cars (see Supplementary Fig. 2a). Besides, the decreasing rate has a limit value of roughly 12% according to the fitting line in Fig. 1c. This limit reveals the maximum benefit in emission reduction that can be achieved through truck platooning.

### Truck platooning degrades road infrastructure faster

Although truck platooning is beneficial in reducing vehicle emissions, it alters the loading intervals between trucks on road pavements and potentially affects their damage growth rates. We developed damage models to evaluate the effects of truck platooning vs. non-platooning on road durability (See Supplementary Method 4). Commonly observed damages on road pavements can be divided into two groups: permanent deformation (also known as rutting) and cracking-related distress (fatigue cracking, potholes, etc.)[30,31]. The LTPP database

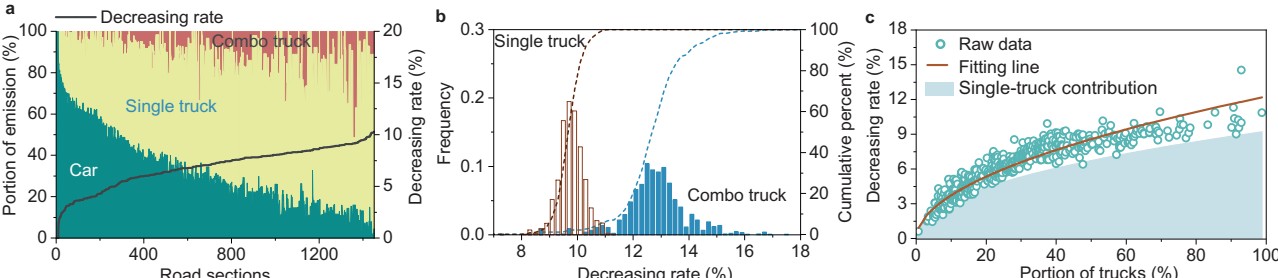

**Fig. 1 | GHG emissions associated with vehicle operations. a** Contributions by passenger cars, single trucks, and combo trucks at the 1457 road sections, along with their relationships with the overall decreasing rate caused by truck platooning. Three colored blocks represent the portions of emissions, while the black line represents the decreasing rate attributed to platooning. **b** The distributions of decreasing rates of single-truck emissions and combo-truck emissions due to platooning. The histogram refers to the frequencies of the decreasing rates. The

dashed line represents the cumulative percent of the decreasing rates. **c** The overall decreasing rates of vehicle emissions due to truck platooning. The raw data refer to the overall decreasing rates at different road sections. The brown fitting line is derived based on the power function model. The light blue region refers to the decreasing rates of vehicle emissions contributed by single trucks. Source data are provided as a Source Data file.

suggests that cracking-related distresses are the predominant type of pavement failure. At a threshold rutting depth value of 12.7 mm[32], 95% of the measured average rutting depth data at the left and right wheel paths is below this threshold. Therefore, the damage model developed in this research focuses on cracking-related distresses only, although rutting-related damage can also be influenced (even favorably) by truck platooning as reported in existing studies[33–35]. Our findings show that truck platooning results in accelerated road degradation. This degradation effect (DE) is defined as the ratio of road durability under the normal traffic mode to that under the truck platooning mode. A DE value higher than 1.0 indicates that truck platooning reduces the road durability compared to the normal operation mode (i.e., non-platooning), or vice versa.

We find that for over 87% of road sections, the DE varies between 1.0 and 6.0 (Fig. 2a). The average DE reaches 2.1, indicating that the

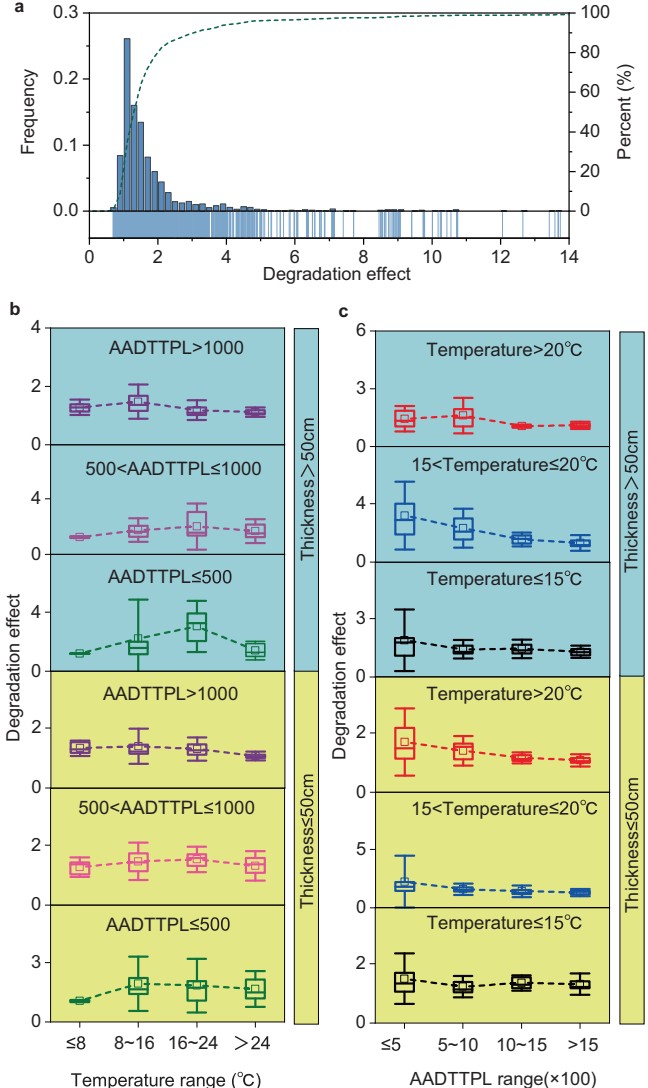

**Fig. 2 | Degradation effects (DEs) of truck platooning on road durability. a** The distributions of DEs on different road sections. The histogram refers to the frequencies of DEs. The dashed line represents the cumulative percent of DEs. **b** The variations of DEs with road temperatures at different scales of road thicknesses and AADTTPLs (annual average daily truck traffic per lane). **c** The variations of DEs with AADTTPLs at different scales of road thicknesses and temperatures. **b**, **c** The error bars refer to ±1.5 times of standard variation (SD) of each group of data. The box-plot elements include the center line, median, and upper and lower quartiles. Source data are provided as a Source Data file.

durability of a large portion of the road infrastructure will be lowered if truck platooning is introduced. Improving road infrastructure durability is always a primary goal for transportation agencies due to the benefits of lowering financial investments and reducing maintenance demands[36–38]. Impaired road durability resulting from truck platooning may hinder the agencies' interest in adopting this technology. The agencies need to balance the benefits of GHG emission reduction by truck platooning with its accelerated damages to road infrastructure.

To find ways to minimize the negative impacts of truck platooning, we assessed relationships between DEs and several potential influencing factors, including road pavement thickness, climatic conditions, and truck volumes. Our results show that climatic conditions (i.e., road temperature) and truck volumes both influence the DEs of truck platooning. DEs are generally higher at intermediate temperatures than those at relatively low or high temperatures (Fig. 2b). DE values decline with the increases in truck volumes (i.e., AADTTPL, annual average daily truck traffic per lane). This is because high truck volumes already make the road approach the truck-platooning situation, i.e., the spacing between trucks becomes very narrow. Consequently, truck platooning does not significantly shift the traffic loading patterns on high-volume roads, leading to small DE values. The plots of DEs on the map (Supplementary Fig. 2b) also show elevated values for urban roads located in the west and east coastal areas of the U.S., where truck volume is relatively low and passenger cars dominate. The above findings suggest that the durability of road infrastructure with low-truck volumes and intermediate temperatures is more easily affected by truck platooning. This situation may change if road infrastructure is better designed to adapt to the platooning situation.

### Truck platooning raises GHG emissions of road infrastructure

Life cycle assessment of road infrastructure emissions includes multi phases, including material production, construction, maintenance, usage, and end-of-life (EOL) processing[39–42]. Emissions from the usage phase refer to vehicle emissions, which have been individually evaluated in this study. We divided the emissions from other phases into two portions to facilitate analysis: emissions from the initial construction stage and those from the road maintenance work after construction (i.e., maintenance, rehabilitation, and reconstruction). Road emissions at the initial construction stage include emissions from material production, material transport, and construction equipment operations. By contrast, road emissions at the maintenance stage are generated from the maintenance material production & transport and maintenance equipment operations. The EOL processing of road materials (i.e., milling and transport) is also considered in the maintenance stage through transport and equipment operation modules. In addition, traffic disruptions due to lane closure during maintenance work also account for the maintenance stage's emissions. Traffic disruptions, including deceleration, acceleration, slowing down and even queuing of vehicles, produce extra emissions than normal vehicle operations. Even though such extra emissions are directly generated from traveling vehicles, they are caused by road maintenance work and thus are assigned to road infrastructure emissions. Road maintenance is closely related to road durability. Truck platooning affects road durability as characterized by the degradation effect (DE) factors (Fig. 1), and it thus changes the maintenance period on the road. Based on DE values, we determined the service life of a road section under truck platooning and proportionally assigned the maintenance period to the road according to the actual maintenance record in the LTPP database. We then assessed emissions from road infrastructure under normal traffic mode and truck-platooning mode (see Supplementary Method 5).

We first analyzed the impacts of maintenance-related traffic disruptions on GHG emissions. Figure 3a shows a comparison of the emissions of vehicles on the 1457 road sections during road maintenance and those during a regular period (i.e., without maintenance).

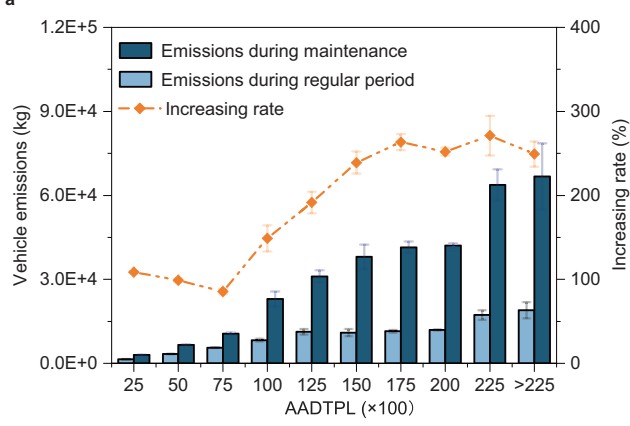

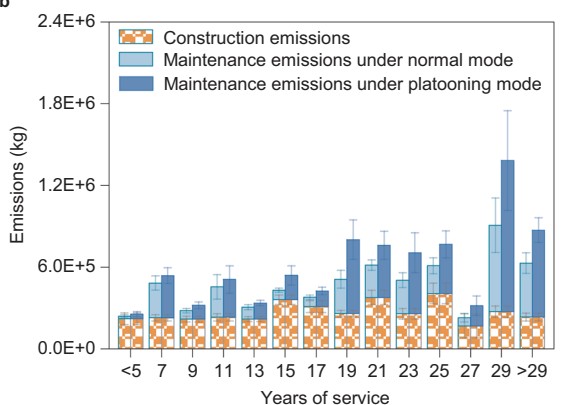

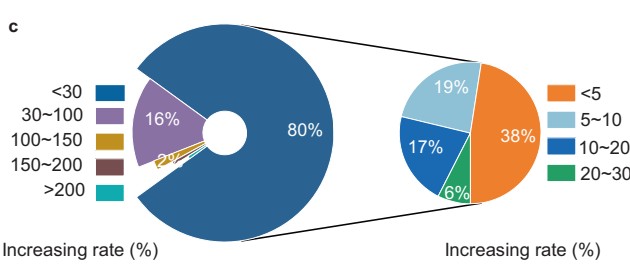

**Fig. 3 | GHG emissions associated with road infrastructure. a** Comparisons of vehicle emissions on 1457 road sections during road maintenance and regular period. The histogram refers to daily vehicle emissions on roads. The dashed line refers to the increasing rates of emissions due to maintenance. The emission value refers to the vehicle emission per driving kilometer. **b** GHG emissions from road infrastructure under normal traffic and truck platooning modes. Data in **b** are grouped according to the service life of road sections. The emission values refer to the averaged emissions of the road sections in each service year group. The emission values represent the emissions from construction or maintenance work on the road with a unit length of 1 kilometer. The error bars in **a** and **b** refer to ±1.5 times of standard error (SE) of each data group. **c** Distributions of the increasing rates of road emissions due to platooning. Source data are provided as a Source Data file.

At the road-network level, we confirm that traffic disruptions associated with maintenance work cause noticeable GHG emissions. The increased emissions become more evident under high traffic volume conditions. As the AADTPLs (annual average daily traffic per lane) exceed 15,000, >200% of additional GHG emissions are generated by traffic disruptions during road maintenance. We further assessed emissions from the maintenance activities and calculated the total emissions from maintenance work (i.e., emissions from maintenance-related traffic disruptions and maintenance activities). We also calculated emissions from the initial road construction for comparison purposes. We observe that

the emissions caused by maintenance work account for 8–54% of the overall road infrastructure emissions (i.e., emissions from the initial construction and from the maintenance work) in North America (Fig. 3b, Supplementary Fig. 3). This indicates that emissions due to road maintenance are non-negligible. A well-planned maintenance strategy is required to reduce road emissions[43–45].

As truck platooning is applied, emissions caused by maintenance work on road infrastructure climb (Fig. 3b). We observe that the maintenance-related emissions increase by at least 21% and even 155% for roads with different service lives (Supplementary Fig. 3). The shares of maintenance-related emissions also rise to 12–66% of the overall road infrastructure emissions. This is attributed to the degradation effect of truck platooning on road infrastructure: truck platooning causes more rapid road degradation, which demands more frequent maintenance work and generates more emissions as compared to the normal traffic mode.

Due to the increase in maintenance-related emissions, the overall emissions from road infrastructure rise. We find that a quintile (20%) of the 1457 road sections emit 30% or more additional GHG emissions if truck platooning is used (Fig. 3c). Specifically, 16% of road sections experience a 30%-100% emission increase, and 2% of road sections experience a 100%-150% rise. In extreme cases, emissions of 2% of road sections rise by 150% or more. For the remaining 80% of the road sections, increases in GHG emissions due to truck platooning are less than 30% but still notable. On average, emissions would increase by 27.9% on all the road sections if truck operations are converted into the platooning mode (Fig. 3c, Supplementary Fig. 2c). As a result, truck platooning can reshape carbon footprints by changing road maintenance schedules. This needs to be considered by transportation agencies in policy formulation. Noteworthy is that road maintenance work improves pavement surface conditions, which helps lower fuel consumption and GHG emissions from vehicles. Therefore, if the road condition-related vehicle emissions are considered, which are treated as part of the vehicle emissions in this research, the impacts of maintenance work on road emissions will be diluted.

**Truck platooning decreases emissions of vehicle-road system**

As truck platooning increases road infrastructure emissions while decreasing vehicle emissions, the tradeoff between the two determines the net decarbonization effect of truck platooning. A comparison of emissions from road infrastructure and those from vehicles indicates that the former is lower than the latter in an integrated vehicle-road infrastructure system (Fig. 4a, b). The gaps between road infrastructure emissions and vehicle emissions become more evident as the road's service year increases. At the initial in-service year, road infrastructure emissions are non-negligible (19% of the total emissions). In the 4th service year, however, vehicle emissions reach 90% of the total emissions, while those from road infrastructure only account for the remaining 10%. This trend reveals that vehicle operations emit more GHG emissions than road construction and maintenance work, especially for roads with relatively long service lives. As truck platooning is applied, the portion of road infrastructure emissions increases to 21% at the beginning year (Fig. 4c), but this influence descends gradually with the rise in the service year and disappears in the 6th year.

As a whole, truck platooning is beneficial for mitigating the total emissions of the vehicle-road system (i.e., vehicle emissions plus road infrastructure emissions). If truck platooning is introduced, for road sections with different service lives, 69–94% of them experience reductions in total emissions, 3%–25% experience emission increases, and 2%–6% of them remain unchanged (Fig. 5a). Supplementary Fig. 2d shows the detailed decreasing rates in total emissions for each road section. We further assessed the cumulative GHG emissions of 1457 road sections from 1987 to 2020. We find that truck platooning would decrease the cumulative GHG emissions from the whole system at a

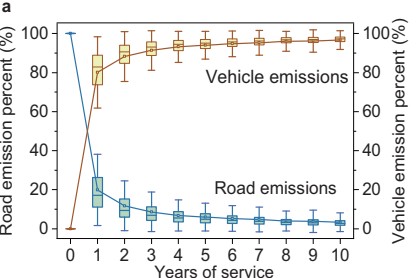 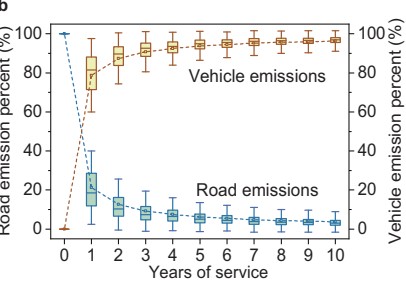 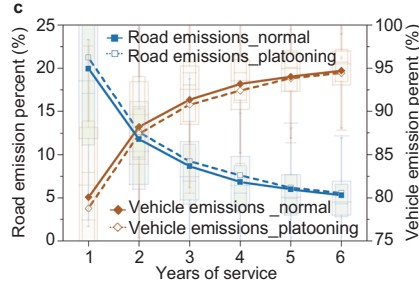

**Fig. 4 | GHG emissions from road infrastructure versus those from vehicles.**
**a** The percentage of road infrastructure emission versus that of vehicle emission under the normal traffic mode. **b** The percentage of road infrastructure emission versus that of vehicle emission under the truck platooning mode. **c** Comparisons of the percentage of road infrastructure emission and that of vehicle emission under the two modes. **a**–**c** The error bars refer to ±1.5 times of SD of each data group. The box-plot elements include the center line, median, and upper and lower quartiles. Source data are provided as a Source Data file.

time horizon of 34 years (Fig. 5b). Clearly, although additional road-phase emissions are generated by truck platooning, they can be fully compensated by reductions in vehicle-phase emissions. Specifically, the annual emissions of road sections decrease by 5.1% on average if truck platooning is applied, corresponding to a reduced amount of ~75 t $CO_2$-eq per kilometer of the road (Fig. 5c). Considering that highway road mileage across North America exceeds 110 thousand kilometers[46], ~8.3 million tons of $CO_2$-eq emissions can be saved per year if truck platooning is introduced to all highways. This is undoubtedly a great benefit for climate change mitigation.

**Truck platooning increases costs of the vehicle-road system**
Besides GHG emissions, truck platooning also affects the costs of the vehicle-road infrastructure system—another crucial factor concerned by transportation agencies and road users. Our analysis shows that truck platooning would increase road infrastructure costs (i.e., Agency costs), with an average increasing rate of 20.8% for 1457 road sections studied (Fig. 6, Supplementary Table 1). The rise in road infrastructure costs is mainly due to more frequent road maintenance work caused by truck platooning. Costs of passenger cars also increase at a moderate rate (3.1% on average), as cars suffer traffic disruptions during road maintenance, generating additional vehicle operating and time delay costs. In contrast, costs of trucks (single truck & combo truck) are lowered by 4.1% on average due to the fuel-saving effect of the platooning mode. Overall, the costs of the vehicle-road system rise by 4.6% on average, equivalent to a rise of $ 16498 per road kilometer (Supplementary Table 1).

## Discussion
In summary, the application of truck platooning needs to be carefully evaluated. Truck platooning leads to a reduction in GHG emissions due to the improved fuel economy of trucks. Still, the beneficial effect varies and depends on the traffic composition of a particular road. However, truck platooning increases the financial burdens of car users and transportation agencies due to more frequent road maintenance work caused by accelerated road deterioration. Therefore, GHG emission reduction by truck platooning comes with a cost. To fully exploit the benefits of truck platooning, roads need to be re-engineered. In particular, road materials and structures need to be strengthened to cope with more demanding loads from platooned trucks. A designated lane may be assigned to platooned trucks to minimize the investments while maximizing the benefits. Findings in this research are expected to provide essential references for formulating managerial and technical strategies to facilitate the broad application of truck platooning.

In addition to the factors considered in this research, there are still many truck platooning-related factors that potentially influence GHG emissions and costs. One of the factors is the influence of truck

platooning on vehicle use and demand. For instance, because the platooning mode reduces fuel consumption and costs of truck operation, this may increase trucks' attractiveness and lead to additional use. Conversely, frequent road maintenance caused by truck platooning increases travel time for cars and trucks, which may lower travel demand. The changes in vehicle use and demand affect road durability and traffic flow state, which further alter emissions from the vehicle-infrastructure system. Policies from the authority can also impact the application of truck platooning. Taxations on platooning trucks may be implemented to compensate for the increasing costs of agency and passenger cars. Guides may also be developed to optimize the truck platooning strategies, such as improving the platooning mode to further decrease road damage[18,47,48] and encouraging the operations of truck platooning at night to reduce road occupancy during the day. All these potential management policies affect the implementation of truck platooning and thus change the emission magnitude. Therefore, the impacts of truck platooning on emissions are rather complicated in the real world. More in-depth research is expected to consider a broader range of influencing factors and evaluate the effects of different policies and strategies.

## Methods
### Road data
Road data are obtained from the LTPP database developed by the US Federal Highway Administration (FHWA). The LTPP database contains approximately 280 million records of road data[23,49]. It is open to the public and is available from the FHWA server (https://infopave.fhwa.dot.gov/Data/DataSelection). From the LTPP database, we selected 1457 road sections which are distributed in 61 states/districts/provinces and 350 counties in the US and Canada. To comprehensively assess the impacts of truck platooning, selected road sections cover different climate regions, functional classifications, and in-service years. The monitoring data of the selected road sections include road site locations, traffic volumes, environmental conditions, road performance, and road maintenance activities. The real-world data enhance the reliability and representativeness of the assessment results. More details on road data acquisition and processing are explained in Supplementary Method 1.

### Traffic flow simulations under normal and platooning modes
Two types of traffic modes are included in this research. One is the normal traffic mode without truck platooning, and the other is the truck platooning mode. Under the truck platooning mode, the rear-ward truck closely follows the preceding one, and the spacing between the adjacent trucks keeps rather narrow and stable. Truck spacing influences the fuel economy and GHG emissions of trucks. It also affects the loading intervals between trucks on road infrastructure. To quantify truck spacings at the two modes, we utilized a traffic flow

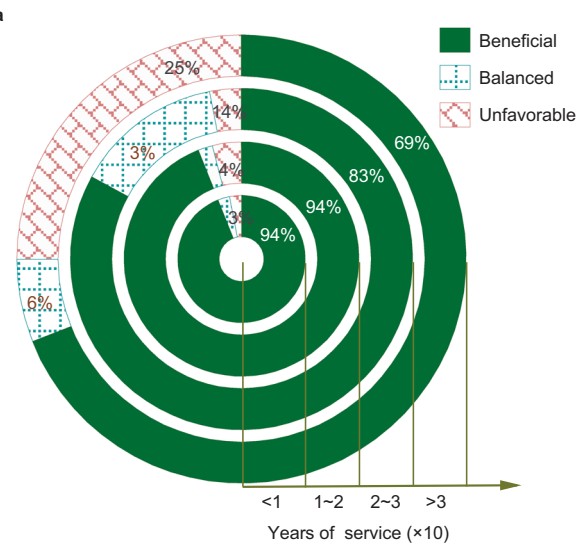

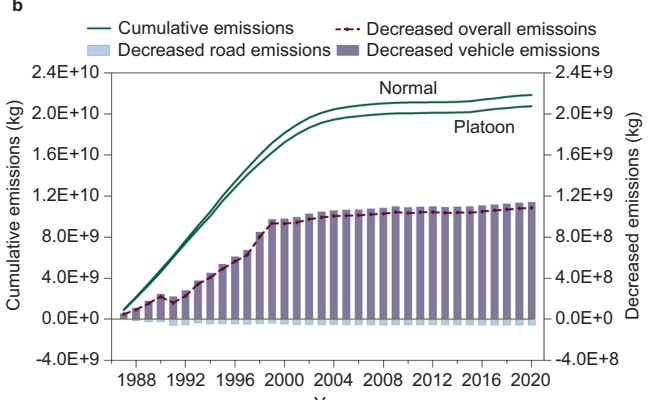

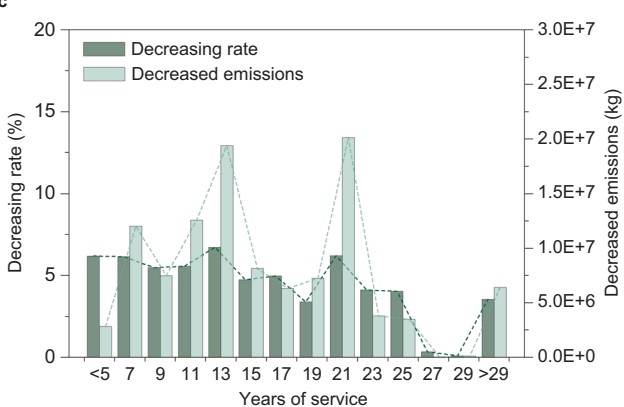

**Fig. 5 | Variations in GHG emission of the integrated vehicle-road system due to truck platooning. a** Impacts of truck platooning on the total vehicle & road emissions of different road sections, which are grouped according to their service lives. The beneficial/unfavorable result represents that truck platooning decreases/increases the total emissions. The balanced result means that truck platooning has almost no effect on the total emissions (i.e., the variation rate of the total emission is within ±0.5%). **b** Cumulative emissions from the vehicle-road infrastructure system at normal traffic mode and truck platooning mode. The solid lines represent the cumulative emissions. The histograms represent the variations in vehicle emissions and road infrastructure emissions due to truck platooning. The dashed line represents the decreased emissions of the vehicle-road infrastructure system due to truck platooning. The cumulative emission slows down after the year of 2000. This is because some road sections have been excluded from monitoring in the LTPP database since 2000. Accordingly, the emissions from those sections are not counted anymore. **c** The decreased amounts and decreasing rates of the annual emissions of road sections due to platooning. Source data are provided as a Source Data file.

headway, $b$ is the desired deceleration, $L$ is the leading vehicle length, and $\Delta v_n(t)$ is the velocity difference between the vehicle $n$ and its preceding vehicle $n-1$, which is calculated as follows.

$$\Delta v_n(t) = v_{n-1}(t) - v_n(t) \tag{2}$$

The input parameters of the improved IDM were calibrated using the Next Generation SIMulation (NGSIM) trajectory data collected at the Hollywood Freeway (U.S.101) and Berkeley Highway (I-80) in California[50]. The adopted parameters and models are deemed realistic to capture the traffic flow characteristics on US roads.

The truck platooning mode in the framework was simulated with a cruising controller and a gap-regulating controller[26]. The cruising controller maintains the user-desired speed when the preceding vehicle is absent or far away. The acceleration of a cruising vehicle is modeled as:

$$a_{n,k} = k_0 \cdot (v_{set} - v_{n,k-1}) \tag{3}$$

Where, the control gain $k_0$ is a parameter to determine the rate of speed error for acceleration, $v_{set}$ is the driver's desired speed and $v_{n,k-1}$ is the speed of vehicle $n$ at time step $k$. The value of $k_0$ is assumed as 0.4 s$^{-1}$ according to reference[52].

In the gap-regulating mode, the car-following response of the first truck in the platoon is described by:

$$a_{n,k} = k_1 \cdot e_{n,k} + k_2 \cdot (v_{n-1,k-1} - v_{n,k-1}) \tag{4}$$

Where, $e_{n,k}$ is the gap error of vehicle $n$ at time step $k$. An existing study found that the vehicle acceleration depends on the gap error and the speed difference with the preceding vehicle, where their feedback gains $k_1$ and $k_2$ are 0.23 s$^{-2}$ and 0.07 s$^{-1}$, respectively[53].

For the following trucks in the platoon, their speeds are calculated by the speed in a previous time step $v_{n,k-1}$, the gap error $e_{n,k-1}$ in a previous time step and the corresponding derivative. Equation (5) is used for this calculation.

$$v_{n,k} = v_{n,k-1} + k_p \cdot e_{n,k-1} + k_d \cdot \dot{e}_{n,k} \tag{5}$$

Where, $k_p$ and $k_d$ are determined as 0.45 s$^{-1}$ and 0.25, respectively[53,54]. The gap error $e_{n,k-1}$ is calculated by Eqs. (6) and (7).

$$e_{n,k-1} = x_{n-1,k-1} - x_{n,k-1} - L - t_{des} \cdot v_{n,k-1} - d_0 \tag{6}$$

$$d_0 = \begin{cases} 0, v \geq 10 m/s \\ -0.125v + 1.25, v < 10 m/s \end{cases} \tag{7}$$

simulation framework to determine the spatio-temporal distributions of vehicles (trucks & passenger cars) on the road. The driving behaviors of vehicles under the normal operation mode are described by an improved Intelligent Driver Model (IDM)[50,51], which is shown as follows.

$$\begin{cases} \dfrac{d^2 x_n(t)}{dt^2} = a_n \left[ 1 - \left( \dfrac{v_n(t)}{V_n} \right)^{\delta_n} - \left( \dfrac{S_n(v_n(t), \Delta v_n(t))}{\Delta x_n(t) - L_n} \right)^2 \right] \\ S_n(v_n(t), \Delta v_n(t)) = s_{n,0} + s_{n,1} \cdot \sqrt{\dfrac{v_n(t)}{V_n}} + \tau_n v_n(t) - \dfrac{v_n(t) \cdot \Delta v_n(t)}{2\sqrt{a_n b_n}} \end{cases} \tag{1}$$

Where, $a$ is the maximum acceleration, $v$ is the actual velocity, $V$ is the desired velocity, $\delta$ is the acceleration exponent, $S(\cdot)$ is the desired minimum gap, $s_{n,0}$ and $s_{n,1}$ are the jam distances, $\tau$ is the safe time

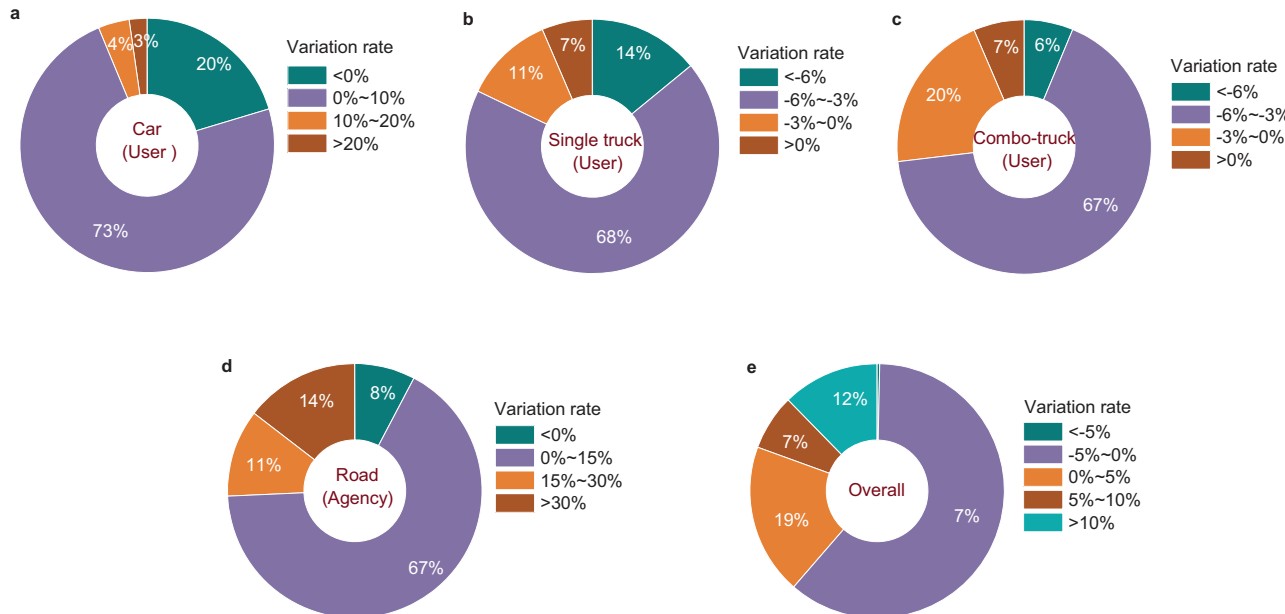

**Fig. 6 | Variations in costs of 1457 road sections due to truck platooning. a** The distribution of the increasing rates of car costs. **b** The distribution of the variation rates of single-truck costs. **c** The distribution of the variation rates of combo-truck costs. **d** The distribution of the variation rates of road infrastructure costs. **e** The distribution of the variation rates of overall costs (vehicle costs & road costs). **a–e** A positive rate (i.e., >0%) indicates an increase in the cost, while a negative rate indicates a decrease in the cost. Source data are provided as a Source Data file.

Where, $x_{n-1,k-1} - x_{n,k-1}$ is the inter-truck spacing, $t_{des}$ is the desired time gap, $L$ is the vehicle length, and $d_0$ is the spacing margin[55]. More details on the traffic flow simulation framework are available in Supplementary Method 2.

### Fuel consumption model of trucks under the platooning mode

The aerodynamics of a truck running in a platooning mode differ from those of a separately running truck. The lead truck in a platoon resists most of the drag resistance, while the trailing ones experience less air resistance[12,56,57]. The reductions in drag resistances lower the required work from truck engines[9,11,16,58]. Accordingly, fuel consumption and GHG emissions of trucks in the platooning mode are reduced. We evaluated the aerodynamic effects of truck platooning based on the Computational Fluid Dynamics (CFD) simulation method. In the simulation, trucks in a platoon are aligned in a straight line to improve their aerodynamic efficiency. Based on the aerodynamic simulations, we developed a model to assess the fuel-saving rate ($\Delta FC$) of truck platooning, as shown in Eq. (8).

$$\Delta FC = \frac{0.433 \cdot e^{0.008 \cdot S} \cdot [a \cdot \ln(S) + b] \cdot L^c}{1 + \frac{r_0 mg}{\frac{1}{2}\rho v^2 A \cdot (0.014 \cdot L + 0.366)}} \quad (8)$$

Where, $S$ is the separation distance, $L$ is the truck length, $r_0$ is the coefficient of road resistance, $m$ is the mass of the truck, $\rho$ is the density of the air, $v$ is the velocity of the truck relative to the air fluid, $A$ is the front area of the truck, and $a$, $b$ and $c$ are coefficients related to the truck position in a platoon.

We further calibrated the above model with the fuel consumption data monitored in the Partially Automated Truck Platooning (PATP) project[11]. The PATP project tested the fuel economy of a three-vehicle truck platooning system. After calibration, the model is reliable in estimating the fuel-saving effect of truck platooning. Based on the fuel-saving rate, the fuel consumption of the truck is eventually determined using Eq. (9).

$$FC_P = FC_0 \cdot (1 - \Delta FC) \quad (9)$$

Where, $FC_P$ is the fuel consumption of the truck in a platoon, $FC_0$ is the original fuel consumption of the truck, and $\Delta FC$ is the saving rate of fuel consumption due to platooning. With Eqs. (8) and (9), the truck's fuel consumption is calculated and then used to assess its GHG emissions. More details on developing fuel consumption models are available in Supplementary Method 3.

### Evaluation of road deterioration caused by truck platooning

Roads are continuously damaged by repeated truck loads. As a result, road maintenance work is needed after several years of service. The maintenance period is directly dependent on the damaged state of the road. The damage accumulation of the road infrastructure is influenced by the loading interval (i.e., rest period). For cracking- and fracture-related damages, a longer rest period generates a more pronounced healing effect on road pavement material than a shorter one, and thus results in less damage accumulation. Truck platooning triggers high-frequency loading repetitions with short rest periods as compared to normal traffic loads. As a result, it accelerates the damage accumulation of the road and thus affects its maintenance period. We developed a damage model to assess road durability under truck platooning and normal traffic modes. The damage model consists of three sub-models, i.e., the fatigue life prediction model, road response model, and damage accumulation model.

The fatigue life prediction model, as shown in Eq. (10), was established based on both laboratory and field tests on road materials and structures (see Supplementary Method 4). Analysis methods, including the dissipated energy method and the viscoelastic continuum damage (VECD) method, were used to fit the test results and develop the fatigue life prediction model. The model includes the effects of road response, climatic conditions, and rest period (i.e., loading interval between two loads) on the fatigue resistance of the road. Road response required in the fatigue life prediction model is estimated based on the mechanical analysis of road structure, as shown in Eq. (10). The road response model also considers various factors influencing road durability, including road structures, material properties, climatic conditions and axle configurations. The effects of wheel wander distributions of trucks are also included in the road

response model, as the platooning trucks may have different lateral offsets compared with human-driven trucks[47,48,59], due to the superior control ability of the platooning technology. The lateral offsets of non-platooning trucks (i.e., human-driven trucks) are assumed to follow a normal distribution, while those of platooning trucks are designed to distribute evenly across the wheel path to lower the truck's damaging impacts. Noteworthy is that trucks within a platoon follow the same driving path to ensure their aerodynamic efficiency, while trucks in different platoons are assigned to load evenly across the wheel path. More details on the fatigue life prediction model and road response model are available in Supplementary Method 4.

$$\begin{cases} N_f = a \cdot \varepsilon^{-b} \cdot E^{-c} \cdot e^{(d \cdot T + f \cdot RP)} \cdot SF \\ \varepsilon = G[h(x), E(x), v(x), T(x), AL] \end{cases} \quad (10)$$

Where, $N_f$ is fatigue life, $\varepsilon$ is the strain level of road structure caused by traffic load, $T$ is road temperature, $E$ is the initial stiffness modulus of a particular layer in the road structure, $RP$ is the rest period between two traffic loads, $SF$ is a shift factor connecting the laboratory and field fatigue life, $G$ is a road response function, $x$ is the particular layer of the road, and $a, b, c, d, f$ are model parameters that can be found in Supplementary Method 4. $h(x)$, $E(x)$, $v(x)$, and $T(x)$ are the thickness, stiffness modulus, Poisson's ratio, and temperature of layer $x$, respectively. $AL$ is the axle load information, including truck classification, axle configuration, gross weight, wheel wander and tire contact area.

Combined with the fatigue life prediction model and the road response model, the damage accumulation model was established to evaluate the accumulative damage on the road infrastructure induced by traffic loadings, as shown in Eq. (11). With Eq. (11), damage evolutions of a road under the normal and truck platooning modes are determined and compared. The comparison results are used to evaluate the deterioration effect of truck platooning on the road and proportionally assign maintenance work on road infrastructure. More details on the damage model development and deterioration effect assessment are available in Supplementary Method 4. Noteworthy is that Eq. (11) mainly focuses on evaluating the fatigue damage caused by truck loads on road pavement.

$$D = H(N_{f,i}, \varepsilon_i, n_i) \quad (11)$$

Where, $D$ is the damage extent of road, $H$ is the damage function, $N_{f,i}$ is the fatigue life of the road under the $i$-th traffic loading, $\varepsilon_i$ is the road response under the $i$-th traffic loading, and $n_i$ is the total number of the $i$-th traffic loading.

### GHG emission and cost models for vehicle-road system

GHG emissions of the integrated vehicle-road infrastructure system consist of emissions from both road infrastructure and vehicle operations. As mentioned, road infrastructure emissions are generated from the initial road construction, the road maintenance, the EOL processing and the maintenance-related traffic disruptions. Therefore, we used Eqs. (12)–(14) to calculate road infrastructure emissions ($GHG_{RI}$). The quantity of the construction, maintenance or EOL processing activity was calculated based on road dimensions, while traffic disruption states during road work were estimated using the RealCost software[60]. The detailed emission intensity values can be found in Supplementary Method 5.

$$GHG_{RI} = GHG_{CME} + GHG_{TD} \quad (12)$$

$$GHG_{CME} = \sum_{i=1}^{n} f_{CMEi} \cdot q_{CMEi} \quad (13)$$

$$GHG_{TD} = \sum_{j=1}^{m} VMT_{TDj} \cdot f_{TDj} - VMT_{Nj} \cdot f_{Nj} \quad (14)$$

Where, $GHG_{CME}$ is the emission from the road construction, road maintenance or EOL processing activities, including the material production, material transport, and equipment operations. $GHG_{TD}$ is the additional emission caused by traffic disruptions during road maintenance work. $f_{CMEi}$ is the unit emission intensity of the $i$-th construction, maintenance, or EOL processing activity (i.e., production, transport or equipment operation). $q_{CMEi}$ is the quantity of the $i$-th activity. $VMT_{TDj}$ is the vehicle miles traveled at the $j$-th traffic disruption state (deceleration/acceleration, slowing down, queuing). $f_{TDj}$ is the unit emission intensity at the $j$-th traffic disruption state. $VMT_{Nj}$ is the vehicle miles traveled at a normal traffic state (no maintenance). $f_{Nj}$ is the unit emission intensity at a normal traffic state.

The vehicle-phase emissions refer to the GHG emissions generated by vehicles traveling on the road section without being interfered by road maintenance work. Such portions of emissions are estimated based on the vehicles' fuel consumption. In the truck platooning mode, the fuel consumption of trucks is saved due to reduced air resistance (see Eq. (8)). Such savings are considered in evaluating the vehicle's emissions. In addition, vehicle emissions are closely related to road roughness (defined as the international roughness index, IRI)[61–63]. The high roughness of a terrible road commonly drives up fuel consumption and emissions. Therefore, road performance data (i.e., IRI) from the LTPP database was incorporated into the vehicle emission model. Eventually, we developed the model to estimate vehicle emissions ($GHG_V$), as expressed by Eq. (15).

$$GHG_V = \sum_{i=1}^{n} f_{Vi} \cdot VMT_i \cdot FC_i \cdot (1 - \Delta FC_{Pi}) \cdot (1 + \Delta FC_{IRIi} \cdot \Delta IRI) \quad (15)$$

Where, $GHG_V$ is emissions from vehicles, $f_{Vi}$ is the unit emission intensity of the $i$-th vehicle, $VMT$ is the distance traveled by the $i$-th vehicle, $FC_i$ is the fuel consumption of the $i$-th vehicle, and $\Delta FC_{Pi}$ is the saving rate due to truck platooning. If the target vehicle is a passenger car, $\Delta FC_{Pi}$ is assigned to be 0%. Otherwise, $\Delta FC_{Pi}$ is calculated using Eq. (8). $\Delta FC_{IRIi}$ is the variation rates of vehicle fuel consumption due to road's IRI. $\Delta IRI$ is the gap between the actual IRI of the road and the baseline IRI.

Based on Eqs. (12)–(15), GHG emissions from the road phase and vehicle phase are calculated, and their sum is the total emissions of the integrated vehicle-road system. The cost model for the vehicle-road system has a similar formulation as the emission model, except that the emission intensity in the equation is replaced with the cost intensity. In addition, two more cost components related to tire wear-and-tear and vehicle repair are included in the cost model compared to the emission model[64]. The elaboration of the cost model is not introduced here to save paper space. More details regarding the developments of the emission models and cost models are available in Supplementary Method 5.

## Data availability

The LTPP database is available from the FHWA server (https://infopave.fhwa.dot.gov/Data/DataSelection). The road and traffic data used in this study are provided in the Supplementary Data. Source data are provided with this paper.

## Code availability

The RealCost software and the user's manual are available from the FHWA website (https://www.fhwa.dot.gov/infrastructure/asstmgmt/lccasoft.cfm).

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

## Acknowledgements

The research was supported by grants from the National Natural Science Foundation of China (Grant No. 52108412, Recipient: H.C.), the Research Grant Council of Hong Kong Special Administrative Region Government (Grant No. 15208518 & 15210321, Recipient: Y.W.) and the National Key R&D Program of China (Grant No. 2018YFB1600100, Recipient: L.S.). The sponsorships are gratefully acknowledged. The authors appreciate the data from the LTPP online database, which supports the analysis of the research. The authors also appreciate the assistance provided by Dr. Chang Lu and Dr. Siqi Jia in designing traffic models and maps.

## Author contributions

H.C., Y.W., and L.S. conceived the study. L.S. proposed the conceptual and analytical framework. H.C., Y.W., D.C., and C.X. designed the methods. H.C., Y.W., R.Y., and T.J. performed the analysis. H.C. and Y.W. wrote the manuscript. J.L., K.G., and L.S. contributed to improving the manuscript. All of the authors discussed the results and reviewed the manuscript.

## Competing interests

The authors declare no competing interests.
