## [Peer Review File · Nature Communications]

Truck platooning reshapes greenhouse gas emissions of the integrated vehicle-road infrastructure systemREVIEWER COMMENTS

Reviewer #1 (Remarks to the Author):

The study is very well thought out, the gaps in the literature are identified clearly and the methodology is very clearly explained. It was easy to understand and follow. The supplementary materials are very detailed, showing every calculation step and how the results are obtained. However, I have some comments/questions on the impact of platooning on infrastructure as that is my main area of research. Please see below

Page 2 line 49 – Platooning trucks cause high-frequency loading repetitions which affect load durability compared to normal operation and consequently increase the demand on road maintenance after initial construction –

I think this sentence is a little too generalized and I would like to bring some other points of view to the author's attention.

First, I am assuming that, even though it is not explicitly stated that the authors assume that the platooning vehicles are also channelized meaning that the following truck loads exactly the same point with the previous truck (meaning they are also autonomous). The reason I assume that is because the fatigue prediction model has a rest period correction, but that is only accurate for the same strain level (same wheel position). While it is true that fatigue life may diminish, some research actually show that rutting life may increase under channelized traffic with reduced rest period. Please refer to the studies "Impact of rest period on asphalt concrete permanent deformation" and "Truck-Platooning Impacts on Flexible Pavements". You can also see similar trends in the study "Development of Stress Sweep Rutting (SSR) test for permanent deformation characterization of asphalt mixture" which explores rutting tests and happens to come across the same finding. So in channelized traffic mode, the impact of rest period may be a little more complicated.

If the authors did not assume channelized traffic, but human driven or even better, distributed traffic, then truck platooning may actually increase the pavement life if the position of the vehicles are adjusted accordingly. The authors already used the "All for one: Centralized optimization of truck platoons to improve roadway infrastructure sustainability" citation, which makes the same point. There are other studies such as "Impact of Autonomous and Human-Driven Trucks on Flexible Pavement Design", "Optimization of Truck Platoon Wander Patterns Based on Thermo-Viscoelastic Simulations to Mitigate the Damage Effects on Road Structures" and "Optimization of Platoons' Lateral Position Based on Damage Estimation" which shows that by distributing the traffic across the lane, the negative impacts of channelized traffic may be reduced drastically, even improving the sustainability of road construction overall which is an important consideration. So I think it is worth mentioning the channelization assumption in this study or correct for some of the wheel wander impacts.

Chapters truck platooning degrades road infrastructure faster and truck platooning increases GHG emissions

In this chapter, I like the use of degradation effect DE index. I think it is a nice metric to show the impact of platooning. However, as I explained in the previous bullet point, I am not sure if we can conclude that truck platooning degrades road infrastructure faster just by looking at fatigue and assuming channelized traffic. I think there is a more nuanced relationship. Therefore, either these assumptions should be explicitly stated or more types wheel position and pavement damage should be consider.

On page Page 7 line 174, it is stated that road infrastructure emissions include those from initial construction stage and maintenance work after construction

However, there are more stages to the pavement life cycle such as material production, construction, maintenance, use and end of life. In this study, the authors separated the use stage

as a separate component and neglected the end of life, which is okay, but I think this should be explicitly stated as generally, use stage or vehicle emissions are also a part of the pavement life-cycle as stated in FHWA website sustainable pavement program.

Regarding the use stage, the authors considered maintenance disruptions to traffic, which is indeed one of the significant sources of delay and emissions. However, another large source of emissions is during normal operation due to rough pavement condition, which increases fuel consumption. You can see these findings at "Estimating the Effects of Pavement Condition on Vehicle Operating Costs", "Vehicle energy consumption and an environmental impact calculation model for the transportation infrastructure systems". While it is true that maintenance increases emissions due to disruptions, because it actually improves pavement surface, it may actually also reduce normal operation costs and emissions, as shown in "Economic and Environmental Impacts of Platoon Trucks on Pavements". So while it is true that maintenance frequency may increase emissions, there may be a tradeoff due to improved pavement surface conditions. That being said, it is true that if use stage is excluded, or looked at separately, the authors conclusion is still valid. So while the authors do make this comparison later (use stage emissions vs the remaining stages being 90% and 10%), I think this additional operation emission part may change some of the trends that are found.

On page Page 13- line 307, when talking about the increase in cost, it is important to consider what I mentioned in the previous bullet point. While the passenger vehicles may have slightly increased cost due to maintenance, that may be offset due to improved pavement condition and overall less fuel consumption. There are similar studies showing this trend such as "Effects of Pavement Condition on LCCA User Costs".

Overall, while making the statement the cost of vehicle road-system rise by 30.4%, again, this excludes the savings due to improve road condition and assume channelized traffic which are the worst case scenarios. I think it is important to mention this in the study if additional analyses are not carried out.

Overall, I think the study has valuable insights and information, however, some of the assumptions either should be explicitly stated multiple times or additional analyses should be conducted to look at the impact of platooning when vehicles are not channelized and when rutting and normal operation is also considered.

Reviewer #2 (Remarks to the Author):

This is a well-written paper on a timely topic, adding to the literature on the impact of truck platooning on emissions. I have some more general comments and some detailed comments.

General comments

- The paper would benefit from first explaining all factors that influence emissions due to the implementation of truck platooning, at least as included in your calculations. Maybe linked in a conceptual model?
- I guess the additional degrading results from all trucks driving on the same 'lanes'. Can this not be easily avoided by applying a random distribution of where trucks drive, within or between platoons?
- Explain more explicit if and how emissions of other road vehicles are affect.
- Explain more clearly factors that might influence emissions but that you did not include. For example (or: at least): if truck platooning reduces fuel use, this reduces costs, leading to additional truck use. And additional maintenance indeed increases travel times (cars, trucks), leading to lower demand.
- I am not sure if it appropriate for this journal, but some more reflection on the policy relevance would be appreciated.

Details

- 104: explain how this value of 14% can be higher than for trucks only. That would mean other

road vehicles (mainly cars) would have to reduce their emissions even more.

- 136: Explain how to interpret the values of the DE-effect.
- The label of Figure 3 is not very clear. What exactly do we see here?

Reviewer #3 (Remarks to the Author):

The work is very interesting and generates important results in the transport engineering area. The logistics planning of transport systems involves many areas, including the operation of vehicles and the operating conditions and maintenance of highways. The concern with energy efficiency in order to reduce fuel consumption and consequently avoid greenhouse gas emissions is a concern of society, in this regard, studies are mostly focused on vehicle operation, fuel consumption, and the life cycle of vehicles and fuels.

This research performs the analysis of emissions considering the impact of operating trucks platooning on pavements and the emissions arising from the need for more frequent maintenance of pavements. This is a new and very important approach when evaluating a broader system, comparing on the one hand the emissions avoided in vehicle operation and on the other, the emissions resulting from the need to increase the frequency of road maintenance. Thus, in addition to the study bringing relevant results, it shows the importance of a systemic analysis considering not only the gains of an operational measure but its impact on the entire system necessary for heavy vehicle traffic.

The developed methodology contains several stages and many simulations. The authors presented additional material describing all steps of the methodology, data used, equations, simulations performed, and results. The analyses were based on a wide and diverse set of data, obtained from different sources, all of them reliable. The data were made available in a set of 5 worksheets. The information provided will undoubtedly allow the reproduction of the work.

The analyses of the results are validated by the results of the simulations and endorse the conclusions of the article. The text is well written and structured, with no difficulty in reading or understanding it.

In my understanding, this work brings important results to the scientific community. I just have a comment for the authors in explaining which greenhouse gases were considered in the analyses, would it be CO₂ equivalent?

Profa. Dra. Dominique Mouette
University of São Paulo

Authors' responses to the reviewers' comments

No. NCOMMS-23-10711 submitted to *Nature Communications*

Title: Truck platooning reshapes GHG emissions of the integrated vehicle-road infrastructure system

Author: Huailei Cheng, Yuhong Wang*, Dan Chong, Chao Xia, Lijun Sun*, Jenny Liu, Kun Gao, Ruikang Yang, Tian Jin

June, 2023

We thank the reviewers and the editor for your relevant and useful comments. In this document, the reviewers' comments are shown in bold fonts, and our replies follow in ordinary print.

1 Responses to Reviewer #1

1.1 The study is very well thought out, the gaps in the literature are identified clearly and the methodology is very clearly explained. It was easy to understand and follow. The supplementary materials are very detailed, showing every calculation step and how the results are obtained.

Thank you so much for reviewing this paper. We also appreciate your encouraging comments on the paper.

1.2. Page 2 line 49 – Platooning trucks cause high-frequency loading repetitions which affect load durability compared to normal operation and consequently increase the demand on road maintenance after initial construction –

I think this sentence is a little too generalized and I would like to bring some other points of view to the author's attention.

First, I am assuming that, even though it is not explicitly stated that the authors assume that the platooning vehicles are also channelized meaning that the following truck loads exactly the same point with the previous truck (meaning they are also autonomous). The reason I assume that is because the fatigue prediction model has a rest period correction, but that is only accurate for the same strain level (same wheel position). While it is true that fatigue life may diminish, some research actually show that rutting life may increase under channelized traffic with reduced rest period. Please refer to the studies "Impact of rest period on asphalt concrete permanent deformation" and "Truck-Platooning Impacts on Flexible Pavements". You can also see similar trends in the study "Development of Stress Sweep Rutting (SSR) test for permanent deformation characterization of asphalt mixture" which explores

rutting tests and happens to come across the same finding. So in channelized traffic mode, the impact of rest period may be a little more complicated.

If the authors did not assume channelized traffic, but human driven or even better, distributed traffic, then truck platooning may actually increase the pavement life if the position of the vehicles are adjusted accordingly. The authors already used the “All for one: Centralized optimization of truck platoons to improve roadway infrastructure sustainability” citation, which makes the same point. There are other studies such as “Impact of Autonomous and Human-Driven Trucks on Flexible Pavement Design”, “Optimization of Truck Platoon Wander Patterns Based on Thermo-Viscoelastic Simulations to Mitigate the Damage Effects on Road Structures” and “Optimization of Platoons’ Lateral Position Based on Damage Estimation” which shows that by distributing the traffic across the lane, the negative impacts of channelized traffic may be reduced drastically, even improving the sustainability of road construction overall which is an important consideration. So I think it is worth mentioning the channelization assumption in this study or correct for some of the wheel wander impacts.

Thank you so much for your insightful comments on the paper. In the original paper, we assumed that both platooning and non-platooning trucks follow a channelized traffic mode, i.e., the following truck loads at exactly the same point as the previous truck. This assumption did not consider the potential impacts of wheel wander on road damage. After receiving the reviewer’s comments and especially after carefully reading the reference “Impact of Autonomous and Human-Driven Trucks on Flexible Pavement Design” suggested by the reviewer, we have updated our damage model to include the wheel wander effect. For non-platooning trucks (i.e., human-driven trucks), we have adopted a normal distribution to simulate the lateral offsets of trucks on a lane, as recommended by the NCHRP report. For platooning trucks, we consider the following two additional and specially designed distribution modes to lower the damaging impact of truck loads on road pavement:

- Platooning mode 1: Platooning trucks are controlled to load evenly on three sub-lanes of the wheel path to lower the critical strain at the wheel center.
- Platooning mode 2: Platooning trucks are controlled to load evenly across the wheel path to distribute the truck damage at the wheel center.

The schematic diagrams of load distribution modes for non-platooning and platooning trucks are shown as follows.

(a) Normal distribution mode for non-platooning trucks

(b) Mode 1 for platooning trucks

(c) Mode 2 for platooning trucks

Each platooning mode is further divided into two sub-controlling modes:

- Sub mode 1: All platooning trucks are randomly assigned to follow the specially designed distribution mode.
- Sub mode 2: Trucks within a platoon follow the same driving path, while trucks in different platoons are assigned to follow the specially designed distribution mode.

The schematic diagrams of the two sub modes are presented as follows. It is seen that

in sub mode 2, the leading truck in a platoon resists most of the drag resistance, while the trailing ones experience less air resistance. However, in sub mode 1, the trailing trucks may still face noticeable air resistance due to the lateral offset from the driving path of the leading truck.

(a) Sub mode 1

(b) Sub mode 2

Based on these distribution modes, we recalculated the strain responses of the road pavement under non-platooning and platooning truck modes and re-evaluated the degradation effects (DEs) of truck platooning. Noteworthy is that our fatigue models were established based on the fatigue tests that were performed under various combinations of rest periods and strain levels. As a result, the models are deemed applicable to evaluate the damage caused by trucks with different rest periods and lateral offsets (i.e., different strain levels). The updated distributions of DE values are compared with those from our original manuscript in the figure below.

It can be observed that the application of wheel wander distributions (i.e., Platooning mode 1 and Platooning mode 2) results in lower DE values, indicating that the distributed traffic mode indeed helps alleviate the damaging effect of platooning trucks on the road compared to the channelized traffic mode. The combination “Platooning mode 1 + Sub mode 2” generates the lowest DE values as compared with the other types of platooning strategies, implying that it is the most beneficial strategy for reducing fatigue damage caused by platooning trucks on the road pavement. This combination is also the best solution for reducing air resistance and fuel consumption of platooning trucks.

However, the resultant DE values from the optimal distributed traffic mode still exceed 1.0, with the predominant value at 1.2. This reveals that platooning trucks still reduce road durability compared to the normal operation mode (i.e., non-platooning). The damaging effect is attributed to the reduced loading interval between two platooning trucks, which significantly lowers the healing of the pavement material and damages the road pavement more rapidly, even though platooning trucks are distributed more evenly across the wheel path.

The updated DE values from the “Platooning mode 2 + Sub mode 2” strategy (the most beneficial strategy in terms of fatigue damage as well as fuel economy) were then used to analyze the GHG emissions and costs of the road infrastructure, and the updated analysis results are presented in the revised manuscript. We have also included descriptions of the applied distribution mode in both the main manuscript and the supplementary file, as follows:

“Wheel wander, which refers to the lateral offset of a vehicle from the wheel path center, is also an important factor that influences the critical strain of the pavement. The lateral offsets of non-platooning trucks (i.e., human-driven trucks) generally follow a normal distribution, as reported in the NCHRP’s research^[46]. On the other hand, platooning trucks can have their lateral offsets more precisely designed, thanks to the superior control ability of platooning technology. Existing studies have proposed several wheel wander modes for platooning trucks to reduce the damaging impacts of truck loads on the pavement ^[47, 48]. In this research, we considered two of these modes:

- (1) Platooning mode 1: Platooning trucks are controlled to load evenly on three sub-lanes of the wheel path to lower the critical strain at the wheel center.
- (2) Platooning mode 2: Platooning trucks are controlled to load evenly across the wheel path to distribute the truck damage at the wheel center.

The schematic diagrams of the wheel wander distribution modes for non-platooning

and platooning trucks are shown in Fig. 4-10.

(a)

(b)

(c)

Fig. 4- 10. The wheel wander distribution modes for the non-platooning and platooning trucks: (a) the normal distribution mode for non-platooning trucks, (b) the mode 1 for platooning trucks, and (c) the mode 2 for platooning trucks.

Each platooning mode is further divided into two sub-controlling modes:

- Sub mode 1: All platooning trucks are randomly assigned to follow the specially designed distribution mode.
- Sub mode 2: Trucks within a platoon follow the same driving path, while trucks in different platoons are assigned to follow the specially designed distribution mode.

The schematic diagrams of the two sub modes are presented in Fig. 4-11. It is seen that in sub mode 2, the leading truck in a platoon resists most of the drag resistance, while the trailing ones experience less air resistance. However, in sub mode 1, the trailing trucks may still face noticeable air resistance due to the lateral offset from the driving path of the leading truck.

Fig. 4- 11. The schematic diagrams of truck distributions under (a) Sub mode 1 and (b) Sub mode 2.

Based on the distributions described above, the critical strain responses of the road pavement under non-platooning and platooning trucks were calculated. The resulting critical strain values were then used in fatigue life prediction models to estimate the fatigue damage of the road pavement under a specific axle load, along with the shift factor. It was found that the combination “Platooning mode 2 + Sub mode 2” is the most beneficial strategy for reducing fatigue damage caused by platooning trucks on the road. This combination is also the best solution for reducing air resistance and fuel consumption of platooning trucks. Therefore, the “Platooning mode 2 + Sub mode 2” strategy is ultimately adopted as the wheel wander control method for platooning trucks in this research.” (Newly added content in the Supplementary file)

“The road response model also considers various factors influencing road durability, including road structures, material properties, climatic conditions and axle configurations. The effects of wheel wander distributions of trucks are also included in the road response model, as the platooning trucks may have different lateral offsets compared with human-driven trucks^[57-59], due to the superior control ability of the platooning technology. The lateral offsets of non-platooning trucks (i.e., human-driven trucks) are assumed to follow a normal distribution, while those of platooning trucks are designed to distribute evenly across the wheel path to lower the truck’s damaging impacts. Noteworthy is that trucks within a platoon follow the same driving path to ensure their aerodynamic efficiency, while trucks in different platoons

are assigned to load evenly across the wheel path.) (Newly added content in the main file).

As for the rutting-related issue, we agree with the reviewer that truck platooning also affects the permanent deformation of the road pavements. Valuable investigations have been conducted by existing studies, especially those recommended by the reviewer. In this research, we mainly focus on the influences of truck platooning on fatigue damage of the pavement, as this type of distress predominantly affects the overall structure capacity of the road pavement. Our laboratory and field tests were also carefully designed to characterize the fatigue damage caused by the normal or platooning trucks on the road pavement. In addition, the LTPP database reveals that about 95% of the measured rutting depth data of road sections is below the critical rutting threshold (i.e., 12.7mm, see the following figure), suggesting that the road sections investigated in this research are most likely free from rutting-related failure. Therefore, the damage model developed in this research focuses on cracking-related distresses only, although rutting-related damage can also be influenced (even favorably) by truck platooning as reported in existing studies.

We appreciate the reviewer’s reminder of the limitations of our damage model. Therefore, we have added the following sentences to the revised manuscript to explicitly state the model’s limitations:

“Commonly observed damages on road pavements can be divided into two groups: permanent deformation (also known as rutting) and cracking-related distress (fatigue cracking, potholes, etc.)^[30,31]. The LTPP database suggests that cracking-related distresses are the predominant type of pavement failure. At a threshold rutting depth value of 12.7mm^[32], about 95% of the measured average rutting depth data at the left and right wheel paths is below this threshold. Therefore, the damage model developed in this research focuses on cracking-related distresses only, although rutting-related damage can also be influenced (even favorably) by truck platooning as reported in

existing studies^[33-35].”

Moreover, the sentences in the Introduction section, i.e., “Platooning trucks cause high-frequency loading repetitions, which affect road durability compared to normal truck operations ^[14-18] and consequently increase the demands on road maintenance work after initial construction (e.g., crack sealing, patching, milling and overlay)”, have been revised to explain more clearly the impacts of truck platooning on pavement performance. The revised sentences are shown as follows:

“In particular, truck platooning reduces the loading interval between two consecutive truck loads, likely hindering the self-healing of the road pavement layer and damaging road durability as compared to normal truck operations ^[14-18]. Consequently, platooning trucks can increase the demands on road maintenance work (e.g., crack sealing, patching, milling and overlay) after initial construction and shorten the pavement’s service life.”

1.3. Chapters truck platooning degrades road infrastructure faster and truck platooning increases GHG emissions

In this chapter, I like the use of degradation effect DE index. I think it is a nice metric to show the impact of platooning. However, as I explained in the previous bullet point, I am not sure if we can conclude that truck platooning degrades road infrastructure faster just by looking at fatigue and assuming channelized traffic. I think there is a more nuanced relationship. Therefore, either these assumptions should be explicitly stated or more types wheel position and pavement damage should be consider.

Thank you so much for your valuable comments. As stated in Response 1.2, we have updated our damage model to include the impacts of wheel wander, which improves the accuracy of evaluating road pavement damage under platooning and non-platooning modes. Furthermore, we have added explicit descriptions of our model’s limit, that is, neglecting the potential impacts of truck platooning on other types of distresses such as rutting.

1.4. On Page 7 line 174, it is stated that road infrastructure emissions include those from initial construction stage and maintenance work after construction. However, there are more stages to the pavement life cycle such as material production, construction, maintenance, use and end of life. In this study, the authors separated the use stage as a separate component and neglected the end of life, which is okay, but I think this should be explicitly stated as generally, use stage or vehicle emissions are also a part of the pavement life-cycle as stated in FHWA website sustainable pavement program.

Thank you so much for pointing out this problem. We agree with the reviewer that the common pavement life cycle analysis includes multi phases, including material production, construction, maintenance, usage, and end of life (EOL) processing. In our manuscript, we divided the road emissions into two portions to facilitate analysis, i.e., emissions from the initial construction stage and those from the road maintenance work after construction (i.e., maintenance, rehabilitation and reconstruction). The emissions from material production and EOL phases have been implicitly included in those two portions.

The cut-off approach is used for the EOL phase, which assumes that the pavement material will not be recycled after demolition. This is a conservative estimate considering that not all places use recycled pavement material. As a result, the EOL phase in our research focuses on the milling and transport of old road pavement material.

In summary, the road emissions at the initial construction stage include emissions from material production, material transport and construction equipment operations. The road emissions at the maintenance stage are generated from the maintenance material production, material transport, and maintenance equipment operations. The EOL processing of the material (i.e., milling and transport) is also considered in the maintenance stage via the transport and equipment operation modules.

As for the vehicle emissions during the usage stage, we have assessed this portion in the Chapter “Truck platooning decreases GHG emissions from vehicles”. Thank you again for pointing out this problem. By revision, we have supplemented the following sentences in the paper to further clarify the stages we considered in the life cycle assessment of road emissions. We have also modified the method sections in both the main and supplementary files to deliver the information more clearly.

“Life cycle assessment of road infrastructure emissions commonly includes multi phases, including material production, construction, maintenance, usage, and end of life (EOL) processing^[39-42]. Emissions from the usage phase refer to vehicle emissions, which have been individually evaluated in the previous chapter. We divided the emissions from other phases into two portions to facilitate analysis: emissions from the initial construction stage and those from the road maintenance work after construction (i.e., maintenance, rehabilitation and reconstruction). Road emissions at the initial construction stage include emissions from material production, material transport and construction equipment operations. By contrast, road emissions at the maintenance stage are generated from the maintenance material production & transport and maintenance equipment operations. The EOL processing of road materials (i.e., milling and transport) is also considered in the maintenance stage

through transport and equipment operation modules.”

1.5. Regarding the use stage, the authors considered maintenance disruptions to traffic, which is indeed one of the significant sources of delay and emissions. However, another large source of emissions is during normal operation due to rough pavement condition, which increases fuel consumption. You can see these findings at “Estimating the Effects of Pavement Condition on Vehicle Operating Costs”, “Vehicle energy consumption and an environmental impact calculation model for the transportation infrastructure systems”. While it is true that maintenance increases emissions due to disruptions, because it actually improves pavement surface, it may actually also reduce normal operation costs and emissions, as shown in “Economic and Environmental Impacts of Platoon Trucks on Pavements”. So while it is true that maintenance frequency may increase emissions, there may be a tradeoff due to improved pavement surface conditions. That being said, it is true that if use stage is excluded, or looked at separately, the authors conclusion is still valid. So while the authors do make this comparison later (use stage emissions vs the remaining stages being 90% and 10%), I think this additional operation emission part may change some of the trends that are found. When talking about the increase in cost, it is important to consider what I mentioned in the previous bullet point. While the passenger vehicles may have slightly increased cost due to maintenance, that may be offset due to improved pavement condition and overall less fuel consumption. There are similar studies showing this trend such as “Effects of Pavement Condition on LCCA User Costs”. Overall, while making the statement the cost of vehicle road-system rise by 30.4%, again, this excludes the savings due to improve road condition and assume channelized traffic which are the worst case scenarios. I think it is important to mention this in the study if additional analyses are not carried out.

Thank you so much for your comments on the paper. We agree with the reviewer that the pavement surface condition affects the fuel economy and GHG emissions of vehicles. As a result, we have included the IRI index in the vehicle emission model, as expressed by Equation (15).

$$GHG_V = \sum_{i=1}^n f_{Vi} \cdot VMT_i \cdot FC_i \cdot (1 - \Delta FC_{Pi}) \cdot (1 + \Delta FC_{IRI_i}) \times \Delta IRI \quad (15)$$

Where, GHG_V is emissions from vehicles, f_{Vi} is the unit emission intensity of the i -th vehicle, VMT_i is the distance traveled by the i -th vehicle, FC_i is the fuel consumption of the i -th vehicle, and ΔFC_{Pi} is the saving rate due to truck platooning. If the target vehicle is a passenger car, ΔFC_{Pi} is assigned to be 0%. Otherwise,

ΔFC_{P_i} is calculated using Equation (8). ΔFC_{IRI} is the variation rates of vehicle fuel consumption due to road's IRI. ΔIRI is the gap between the actual IRI of the road and the baseline IRI.

We have also considered the benefits of road maintenance work on the pavement surface condition, as shown in Fig. 1-7 from the supplementary file. But as the reviewer pointed out, vehicle emissions from the use stage were looked at separately in this research. Accordingly, the tradeoff due to the improved pavement surface condition was not considered when evaluating the effects of road maintenance work on road emissions. If the vehicle emissions during the usage stage are taken into account, the impacts of maintenance may be diluted. We appreciate the reviewer for pointing out this problem. In the revised paper, we have supplemented the following sentences to explain the apparent impacts of the maintenance on road emissions found in this research.

“Noteworthy is that road maintenance work improves pavement surface conditions, which helps lower fuel consumption and GHG emissions from vehicles. Therefore, if the road condition-related vehicle emissions are considered, which are treated as part of the vehicle emissions in this research, the impacts of maintenance work on road emissions will be diluted.”

(a)

(b)

Fig. 1- 7. The IRI data of the road section: (a) the recorded IRI data, and (b) the IRI

data supplemented by the predicted values.

In the cost analysis, the reduced cost of fuel consumption due to the improved IRI has also been considered in the original paper. But what was neglected is the potential reduction in the tire wear-and-tear costs and repair and maintenance costs of vehicles when IRI is improved. We acknowledge the reference “Effects of pavement condition on LCCA user costs” suggested by the reviewer, which highlighted that a better pavement surface condition causes less damage to vehicle tires and body, leading to a decrease in the cost of tire replacement or vehicle repair. By revision, we have further included these two types of cost components in the calculation, as presented in the following sentences:

“In addition to the cost components shown in Table 5-5, drivers also have to bear the costs of tire wear-and-tear (W&T) and vehicle repair-and-maintenance (R&M). Both these costs depend on the pavement surface condition (i.e., IRI). A better pavement condition generally leads to lower costs. To calculate the W&T and R&M costs, Equations (5.14) and (5.15) were adopted according to the literature [63].

$$Cost_{W\&T} = CI_{W\&T} \cdot [1 + C_{W\&T}(IRI)] \quad (5.14)$$

$$Cost_{R\&M} = CI_{R\&M} \cdot [1 + C_{R\&M}(IRI)] \quad (5.15)$$

Where, $CI_{W\&T}$ and $CI_{R\&M}$ refer to the intensity of two types of cost, and $C_{W\&T}(IRI)$ and $C_{R\&M}(IRI)$ refer to the adjusting functions related to the pavement’s IRI. The exact intensity values and adjusting functions for different vehicles are summarized in Table 5-6.

Table 5- 6. The intensity values and adjust functions for different vehicles.

Items	Passenger car	Single truck	Combo truck
$CI_{W\&T}$	0.0015 \$/km/tire	0.0056 \$/km/tire	0.007 \$/km/tire
$CI_{R\&M}$	0.04 \$/km	0.06 \$/km	0.12 \$/km
$C_{W\&T}(IRI)^1$	$0.0018 \cdot IRI^{2.34}$	$0.0020 \cdot IRI^{2.16}$	$0.0018 \cdot IRI^{2.31}$
$C_{R\&M}(IRI)^1$	$5.1 \times 10^{-5} \cdot e^{1.93 \cdot IRI}$	$1.3 \times 10^{-4} \cdot e^{1.74 \cdot IRI}$	$1.1 \times 10^{-4} \cdot e^{1.91 \cdot IRI}$

1 The unit of the IRI parameter in equations is m/km.”

Regarding the channelized-traffic assumption, we have replaced it with the evenly distributed traffic mode to make full use of the benefits of truck platooning technology.

In addition, we also rechecked the calculation procedure for road and vehicle costs and corrected a coding error. Using the updated analysis methods, we have found that the overall costs of the vehicle-road system increase by 4.5% on average, rather than 30.4% as reported in the original manuscript (the worst scenario—channelized mode). We thank the reviewer again for helping us avoid this mistake.

1.6. Overall, I think the study has valuable insights and information, however, some of the assumptions either should be explicitly stated multiple times or additional analyses should be conducted to look at the impact of platooning when vehicles are not channelized and when rutting and normal operation is also considered.

Thank you again for reviewing this paper and giving us many valuable suggestions. In the revised paper, we have supplemented the explicit explanations of the assumptions included in our analysis. Additionally, we have incorporated the effects of wheel wander in our damage model to improve the accuracy of our analysis results.

2. Responses to Reviewer #2

2.1. This is a well-written paper on a timely topic, adding to the literature on the impact of truck platooning on emissions.

Thank you so much for reviewing this paper. We also appreciate your encouraging comments on the paper.

2.2. The paper would benefit from first explaining all factors that influence emissions due to the implementation of truck platooning, at least as included in your calculations. Maybe linked in a conceptual model?

Thank you so much for your valuable suggestions. To clearly show the factors that influence emissions due to truck platooning, we have supplemented a conceptual diagram in the revised manuscript to present the calculation framework, shown as follows.

Extended Data Fig. 1. The framework for evaluating the impacts of truck platooning on GHG emissions and cost of the integrated vehicle-road infrastructure system.

2.3. I guess the additional degrading results from all trucks driving on the same 'lanes'. Can this not be easily avoided by applying a random distribution of where trucks drive, within or between platoons?

Thank you so much for your comments on the paper. Our original paper assumed that both platooning and non-platooning trucks follow a channelized traffic mode, i.e., the

Editorial Note: Vehicles in the figure above are redacted where no permission to publish could be obtained.

following truck loads at exactly the same point as the previous truck. This assumption neglected the potential impacts of wheel wander on road damage. After receiving the reviewers' comments and reading the reference "Impact of Autonomous and Human-Driven Trucks on Flexible Pavement Design" suggested by Reviewer 1, we have updated our damage model to include the wheel wander effect. For non-platooning trucks (i.e., human-driven trucks), we have adopted a normal distribution to simulate the lateral offsets of trucks on a lane, as recommended by the NCHRP report. For platooning trucks, we consider the following two additional and specially designed distribution modes to lower the damaging impact of truck loads on road pavement:

- Platooning mode 1: Platooning trucks are controlled to load evenly on three sub-lanes of the wheel path to lower the critical strain at the wheel center.
- Platooning mode 2: Platooning trucks are controlled to load evenly across the wheel path to distribute the truck damage at the wheel center.

The schematic diagrams of load distribution modes for non-platooning and platooning trucks are shown as follows.

(a) Normal distribution mode for non-platooning trucks

(b) Mode 1 for platooning trucks

(c) Mode 2 for platooning trucks

Each platooning mode is further divided into two sub-controlling modes:

- Sub mode 1: All platooning trucks are randomly assigned to follow the specially designed distribution mode.
- Sub mode 2: Trucks within a platoon follow the same driving path, while trucks in different platoons are assigned to follow the specially designed distribution mode.

The schematic diagrams of the two sub modes are presented as follows. It is seen that in sub mode 2, the leading truck in a platoon resists most of the drag resistance, while the trailing ones experience less air resistance. However, in sub mode 1, the trailing trucks may still face noticeable air resistance due to the lateral offset from the driving path of the leading truck.

(a) Sub mode 1

(b) Sub mode 2

Based on these distribution modes, we recalculated the strain responses of the road pavement under non-platooning and platooning truck modes and re-evaluated the degradation effects (DEs) of truck platooning. Noteworthy is that our fatigue models were established based on the fatigue tests that were performed under various

combinations of rest periods and strain levels. As a result, the models are deemed applicable to evaluate the damage caused by trucks with different rest periods and lateral offsets (i.e., different strain levels). The updated distributions of DE values are compared with those from our original manuscript in the figure below.

It can be observed that the application of wheel wander distributions (i.e., Platooning mode 1 and Platooning mode 2) results in lower DE values, indicating that the distributed traffic mode indeed helps alleviate the damaging effect of platooning trucks on the road compared to the channelized traffic mode. The combination “Platooning 1 + Sub mode 2” generates the lowest DE values as compared with the other types of platooning strategies, implying that it is the most beneficial strategy for reducing fatigue damage caused by platooning trucks on the road pavement. This combination is also the best solution for reducing air resistance and fuel consumption of platooning trucks.

However, the resultant DE values from the optimal distributed traffic mode still exceed 1.0, with the predominant value at 1.2. This reveals that platooning trucks still reduce road durability compared to the normal operation mode (i.e., non-platooning). The damaging effect is attributed to the reduced loading interval between two platooning trucks, which significantly lowers the healing of the pavement material and damages the road pavement more rapidly, even though platooning trucks are distributed more evenly across the wheel path.

The updated DE values from the “Platooning 2 + Sub mode 2” strategy (the most beneficial strategy in terms of fatigue damage as well as fuel economy) were then used to analyze the GHG emissions and costs of the road infrastructure, and the updated analysis results are presented in the revised manuscript. We have also

included descriptions of the applied distribution mode in both the main manuscript and the supplementary file, as follows:

“Wheel wander, which refers to the lateral offset of a vehicle from the wheel path center, is also an important factor that influences the critical strain of the pavement. The lateral offsets of non-platooning trucks (i.e., human-driven trucks) generally follow a normal distribution, as reported in the NCHRP’s research^[46]. On the other hand, platooning trucks can have their lateral offsets more precisely designed, thanks to the superior control ability of platooning technology. Existing studies have proposed several wheel wander modes for platooning trucks to reduce the damaging impacts of truck loads on the pavement ^[47, 48]. In this research, we considered two of these modes:

- (1) Platooning mode 1: Platooning trucks are controlled to load evenly on three sub-lanes of the wheel path to lower the critical strain at the wheel center.
- (2) Platooning mode 2: Platooning trucks are controlled to load evenly across the wheel path to distribute the truck damage at the wheel center.

The schematic diagrams of the wheel wander distribution modes for non-platooning and platooning trucks are shown in Fig. 4-10.

(a)

(b)

(c)

Fig. 4- 10. The wheel wander distribution modes for the non-platooning and platooning trucks: (a) the normal distribution mode for non-platooning trucks, (b) the mode 1 for platooning trucks, and (c) the mode 2 for platooning trucks.

Each platooning mode is further divided into two sub-controlling modes:

- Sub mode 1: All platooning trucks are randomly assigned to follow the specially designed distribution mode.
- Sub mode 2: Trucks within a platoon follow the same driving path, while trucks in different platoons are assigned to follow the specially designed distribution mode.

The schematic diagrams of the two sub modes are presented in Fig. 4-11. It is seen that in sub mode 2, the leading truck in a platoon resists most of the drag resistance, while the trailing ones experience less air resistance. However, in sub mode 1, the trailing trucks may still face noticeable air resistance due to the lateral offset from the driving path of the leading truck.

(a)

(b)

Fig. 4- 11. The schematic diagrams of truck distributions under (a) Sub mode 1 and (b) Sub mode 2.

Based on the distributions described above, the critical strain responses of the road

pavement under non-platooning and platooning trucks were calculated. The resulting critical strain values were then used in fatigue life prediction models to estimate the fatigue damage of the road pavement under a specific axle load, along with the shift factor. It was found that the combination “Platooning mode 2 + Sub mode 2” is the most beneficial strategy for reducing fatigue damage caused by platooning trucks on the road. This combination is also the best solution for reducing air resistance and fuel consumption of platooning trucks. Therefore, the “Platooning mode 2+ Sub mode 2” strategy is ultimately adopted as the wheel wander control method for platooning trucks in this research.” (Newly added content in the Supplementary file)

“The road response model also considers various factors influencing road durability, including road structures, material properties, climatic conditions and axle configurations. The effects of wheel wander distributions of trucks are also included in the road response model, as the platooning trucks may have different lateral offsets compared with human-driven trucks^[57-59], due to the superior control ability of the platooning technology. The lateral offsets of non-platooning trucks (i.e., human-driven trucks) are assumed to follow a normal distribution, while those of platooning trucks are designed to distribute evenly across the wheel path to lower the truck’s damaging impacts. Noteworthy is that trucks within a platoon follow the same driving path to ensure their aerodynamic efficiency, while trucks in different platoons are assigned to load evenly across the wheel path.) (Newly added content in the main file).

2.4. Explain more explicit if and how emissions of other road vehicles are affected.

Thank you so much for your valuable suggestions. Road sections evaluated in this research are assumed to undergo free traffic flows under normal conditions (i.e., without maintenance activity). This assumption is also supported by traffic simulation results. As a result, GHG emissions of passenger cars under normal conditions are regarded unaffected by truck platooning, considering that the platooning trucks cause negligible disruptions in passenger car operations. However, truck platooning causes more frequent maintenance activities on road sections. During the maintenance period, both passenger cars and trucks are subjected to traffic disruptions, including deceleration, acceleration, slowing down, and even queuing. Due to traffic disruptions, passenger cars produce extra emissions than normal operations. Such extra emissions are assigned to road infrastructure emissions in this research, as they are only generated during maintenance work. As a result, the independent impact of truck platooning on car emissions has not been explicitly revealed. By contrast, it has been included in the effects of truck platooning on road infrastructure emissions (See Fig.

3).

In the revised paper, we have supplemented the following sentences to more clearly clarify the effects of truck platooning on car emissions under normal conditions (i.e., without maintenance activity):

“As for the emissions from passenger cars, they are regarded as unaffected by truck platooning. This is because road sections undergo free traffic flows at normal operation periods (i.e., without maintenance activity) according to our traffic simulations. Thus, the platooning of trucks causes negligible disruptions to passenger car operations.”

2.5. Explain more clearly factors that might influence emissions but that you did not include. For example (or: at least): if truck platooning reduces fuel use, this reduces costs, leading to additional truck use. And additional maintenance indeed increases travel times (cars, trucks), leading to lower demand. I am not sure if it appropriate for this journal, but some more reflection on the policy relevance would be appreciated.

Thank you so much for your valuable suggestions. We agree with the reviewer that there are still many influencing factors on emissions that we did not consider in this research, such as the interactive influences of cost and traveling time on vehicle use and demand, as suggested by the reviewer. In the revised paper, we have supplemented the following sentences to explain the potential influencing factors more clearly. In addition, more discussions on policy relevance have also been supplemented.

“In addition to the factors considered in this research, there are still many truck platooning-related factors that potentially influence GHG emissions and costs. One of the factors is the influence of truck platooning on vehicle use and demand. For instance, because the platooning mode reduces fuel consumption and costs of truck operation, this may increase trucks’ attractiveness and lead to additional use. Conversely, frequent road maintenance caused by truck platooning increases travel time for cars and trucks, which may lower travel demand. The changes in vehicle use and demand affect road durability and traffic flow state, which further alter emissions from the vehicle-infrastructure system. Policies from the authority can also impact the application of truck platooning. Taxations on platooning trucks may be implemented to compensate for the increasing costs of agency and passenger cars. Guides may also be developed to optimize the truck platooning strategies, such as improving the platooning mode to further decrease road damage^[18, 57, 59] and encouraging the operations of truck platooning at night to reduce road occupancy during the day. All

these potential management policies affect the implementation of truck platooning and thus change the emission magnitude. Therefore, the impacts of truck platooning on emissions are rather complicated in the real world. More in-depth research is expected to consider a broader range of influencing factors and evaluate the effects of different policies and strategies.”

2.6. Explain how this value of 14% can be higher than for trucks only. That would mean other road vehicles (mainly cars) would have to reduce their emissions even more.

Thank you so much for your comments on the paper. We can see from Fig. 1 (c) that the decreasing rates of vehicle emissions due to truck platooning are affected by the proportions of trucks on the roads. As expected, more trucks on a road result in higher emission savings if platooning is applied. The decreasing rate has an upper limit value of roughly 12%, where the truck percent is nearly 100% (i.e., “trucks only” as stated by the reviewer). This trend reveals the general law of the decreasing rates of vehicle emissions on 1457 road sections. However, we also note that the decreasing rate is affected by truck type, as the length, front area, and mass of a truck all influence the truck’s fuel-saving rate when in a platoon (see our fuel model, Equation (8)). Fig. 1 (b) clearly shows the differences in the decreasing rates of single trucks and combo trucks. Since the truck types on road sections vary greatly, the decreasing rates can change even if the truck portion keeps the same. This also explains the variations in the decreasing rates plotted in Fig. 1(c). The value 14% is the maximum decreasing rate for road sections evaluated in this research (i.e., upper right point in Fig. 1(c)), which is induced by an extremely high percentage of combo trucks that experience significant fuel savings under the platooning mode. In more extreme situations, the saving rates can be 16% or higher if all trucks on a road section are combo trucks, as seen in Fig. 1(b).

Fig. 1(c)

Fig. 1(b)

Thank you again for pointing out this issue. In the revised paper, we have modified the corresponding sentences to avoid any possible confusion. The revised sentences are shown as follows.

“For most cases, the platooning-caused decreasing rate of the overall vehicle emissions ranges from 0.6% to 12%. The extreme decreasing rate reaches 14%, which is induced by an especially high percentage of combo trucks for which fuel savings under platooning mode are the most significant.”

2.7. Explain how to interpret the values of the DE-effect.

Thank you so much for your valuable comments. The degradation effect (DE) is defined as the ratio of road durability under the normal traffic mode to that under the truck platooning mode. This indicator is used to quantitatively characterize the damaging impact of truck platooning on the road. If the DE value exceeds 1.0, it indicates that truck platooning reduces the road durability compared to the normal operation mode (i.e., non-platooning), and vice versa. We thank the reviewer for pointing out this problem. In the revised paper, we have supplemented the following sentences to interpret the DE values more clearly:

“A DE value higher than 1.0 indicates that truck platooning reduces the road durability compared to the normal operation mode (i. e., non-platooning), or vice versa. “

2.8. The label of Figure 3 is not very clear. What exactly do we see here?

Thank you so much for pointing out this problem. By revision, we have improved the labels of Fig. 3 to enhance its readability. The revised figures are shown as follows.

(a)

(b)

(c)

Fig. 3. GHG emissions associated with road infrastructure. (a), Comparisons of vehicle emissions on the 1457 road sections during the road maintenance and the regular period. The histogram refers to daily vehicle emissions on roads, and the dashed line refers to the increasing rate of emissions due to maintenance. AADTPL

refers to the annual average daily traffic per lane. The emission value refers to the vehicle emission per driving kilometer. (b), GHG emissions from road infrastructure under normal traffic and truck platooning modes. Data in (b) are grouped according to the service life of road sections. The emission values refer to the averaged emissions of the road sections in each service year group. The emission values represent the emissions from construction or maintenance work on the road with a unit length of 1 kilometer. (c), Distributions of the increasing rates of road emissions due to platooning.

3 Responses to Reviewer #3

3.1. The work is very interesting and generates important results in the transport engineering area. The logistics planning of transport systems involves many areas, including the operation of vehicles and the operating conditions and maintenance of highways. The concern with energy efficiency in order to reduce fuel consumption and consequently avoid greenhouse gas emissions is a concern of society, in this regard, studies are mostly focused on vehicle operation, fuel consumption, and the life cycle of vehicles and fuels.

This research performs the analysis of emissions considering the impact of operating trucks platooning on pavements and the emissions arising from the need for more frequent maintenance of pavements. This is a new and very important approach when evaluating a broader system, comparing on the one hand the emissions avoided in vehicle operation and on the other, the emissions resulting from the need to increase the frequency of road maintenance. Thus, in addition to the study bringing relevant results, it shows the importance of a systemic analysis considering not only the gains of an operational measure but its impact on the entire system necessary for heavy vehicle traffic.

The developed methodology contains several stages and many simulations. The authors presented additional material describing all steps of the methodology, data used, equations, simulations performed, and results. The analyses were based on a wide and diverse set of data, obtained from different sources, all of them reliable. The data were made available in a set of 5 worksheets. The information provided will undoubtedly allow the reproduction of the work.

The analyses of the results are validated by the results of the simulations and endorse the conclusions of the article. The text is well written and structured, with no difficulty in reading or understanding it.

In my understanding, this work brings important results to the scientific community. I just have a comment for the authors in explaining which greenhouse gases were considered in the analyses, would it be CO₂ equivalent?

Thank you so much for reviewing this paper. We also appreciate your encouraging comments on the paper. For your question, we confirm that the GHG emission mentioned in the paper refers to the CO₂ equivalent. We also supplemented the explanations in the revised manuscript to clarify the point: “This study examined the effects of truck platooning on the GHG emissions (i.e., CO₂-eq) of the integrated vehicle-road infrastructure system at a network level.”

REVIEWERS' COMMENTS

Reviewer #1 (Remarks to the Author):

Thank you very much for your changes. I have no additional comments or questions. Thank you for your work

Authors' responses to the reviewers' comments

No. NCOMMS-23-10711A submitted to *Nature Communications*

Title: Truck platooning reshapes greenhouse gas emissions of the integrated vehicle-road infrastructure system

Author: Huailei Cheng, Yuhong Wang*, Dan Chong, Chao Xia, Lijun Sun*, Jenny Liu, Kun Gao, Ruikang Yang, Tian Jin

June, 2023

We thank the reviewers and the editor for your relevant and useful comments. In this document, the reviewers' comments are shown in bold fonts, and our replies follow in ordinary print.

1 Responses to Reviewer #1

1.1 Thank you very much for your changes. I have no additional comments or questions. Thank you for your work

Thank you again for reviewing this paper.